# Modeling Corrosion Product Film Formation and Hydrogen Diffusion at the Crack Tip of Austenitic Stainless Steel

**DOI:** 10.3390/ma16175799

**Published:** 2023-08-24

**Authors:** Fuqiang Yang, Jianzhou Zhang, Yue Zhang

**Affiliations:** 1School of Science, Xi’an University of Science and Technology, Xi’an 710054, China; 2College of Engineering, Design and Physical Sciences, Brunel University London, London UB8 3PH, UK; 3School of Mechanical Engineering, Xi’an University of Science and Technology, Xi’an 710054, China; 15991194771@163.com (J.Z.); 20205224132@stu.xust.edu.cn (Y.Z.)

**Keywords:** corrosion product films, hydrogen, diffusion, austenitic stainless steel

## Abstract

Corrosion product films (CPFs) have significant effects on hydrogen permeation and the corrosion process at the crack tip. This paper established a two-dimensional calculation model to simulate the formation of CPFs at the crack tip and its effects on the crack tip stress status and hydrogen diffusion. The CPFs were simplified as a single-layer structure composed of Fe_2_O_3_, the effective CPFs boundary was modeled by the diffusion of oxygen, and the CPF-induced stress was modeled by hygroscopic expansion. The simulation was conducted with two stages; the first stage was to simulate the formation of CPFs formation and its effects on the crack tip stress status, while the second stage focused on the hydrogen diffusion with and without CPF formation under different external tensile loads. The results indicate that the highest compressive stress induced by the formation of CPFs is located at 50~60° of the crack contour and dramatically weakens the crack tip tensile stress at low-stress status. The CPFs can inhibit the hydrogen permeation into the crack tip, and the hydrostatic pressure effects on the redistribution of the permeated hydrogen are significant under larger external load conditions.

## 1. Introduction

Austenitic alloys including Ni-based alloys and austenitic stainless steels (ASSs) are widely used to manufacture structural components in the pressurized water reactors (PWRs) as their good corrosion resistance and mechanical properties. However, environmentally assisted cracking (EAC) can occur in these materials in high-temperature water and undermine the reliability of nuclear power systems [1,2]. It is well known that stainless steels rely on their CPFs, such as a passive film, an oxide film, or a de-alloyed layer, formed on the surfaces in the corrosive environment to resist corrosion. However, the existence of hydrogen has dramatic effects on the EAC [3,4]; much research shows that hydrogen can degrade the CPFs and the mechanical properties of alloys, and lead to unacceptable disaster.

The existence of hydrogen can change the structure of the films and decreases their thickness, which can decrease the stability of passive films [5,6,7]. Luo et al. [5] have found that hydrogen reduces the occurrence of Cr_2_O_3_ and nitrogen in the CPFs, and leads to a reduction in the overall film thickness. Dong et al. [6] found that with a bilayer of film, a continuous decrease in Cr^−^ and increase in Fe^−^ concentrations were observed from the surface of the inner layer to the oxide/substrate interface with increasing dissolved hydrogen (DH) concentration, and this will lead to a weakened protection of corrosion. Jeon et al. [7] found that the film structure changed from a bilayer to three-layer structure with an increasing DH concentration, and the thickness of oxide films decreased simultaneously. The explanation for the reduction in film stability and breakdown when hydrogen is present was the further increase in OH^−^/O^2−^ or O^2−^/Fe^2+^ ratio in the passive film [8,9,10]. Moreover, Zeng et al. [9] contribute it to the additional electric repulsion on Fe^2+^ or Fe^3+^ and oxygen vacancies created by hydrogen, which enhances the diffusion rate of cations and anion vacancies, and explained that the hydrogen will not only decrease the concentrations of O^2−^ and Fe^3+^, but also increase the concentration of Fe^2+^ in the CPFs, resulting in changes in the structure and a decrease in the thickness of the CPFs at a steady state.

Hydrogen can also alter the yield and tensile strength, knocking down the ductility of metals when it exists [11,12]. Many mechanisms and their combination, such as the hydrogen-enhanced decohesion mechanism (HEDE), hydrogen-enhanced local plasticity model (HELP), and adsorption-induced dislocation emission (AIDE) [13,14,15,16,17,18,19,20,21], have been proposed to interpret the degradation of material properties and the formation of crack induced by hydrogen. Even though none of the proposed mechanisms could explain all of the hydrogen embrittlement phenomena, it could be confirmed that the hydrogen diffusion at the crack tip is essential in all the mechanisms to explain hydrogen embrittlement. So, it is worth evaluating the hydrogen transportation at the crack tip.

In the modeling of hydrogen diffusion, many works were established on the framework of the equilibrium relationship between lattice and trapped hydrogen founded by Oriani [22,23], the trapping–detrapping process incorporated model modified by McNabb and Foster [24], the hydrostatic stress term-related model proposed by Sofronis and McMeeking [25], and the further modified model incorporating hydrogen-induced dilatation into the material constitutive equations by Lufrano and Sofronis [26]. Besides the works reviewed by Yang et al. [27], other methods were also mentioned to solve the hydrogen diffusion and trapping, such as molecular dynamics simulation [28,29], first-principles calculation [30], and phase-field models [31]. All these models are proposed to incorporate and explain the effects of the main governing factors such as grain size [32], grain boundary [33,34] dislocation density [35], hydrogen concentration [36], and the external condition such as stress, strain rate, temperature, or corrosive environment [36,37] on hydrogen embrittlement. Moreover, Xu et al. [38] also reviewed the models commonly used over the years in the study of microscopic models of material corrosion mechanisms, and proposed that the development of corrosion big data technology has boosted the theoretical simulation to the data-driven direction, by which the corrosion prediction models could be built by the data mining method to integrate the material corrosion mechanisms and corrosion big data.

However, the corrosion product films (CPFs), which have significant effects on the corrosion process [39,40,41,42], were not included in these models. When a passive film forms on the surface of a crack tip during EAC, even at only several nanometers, it can still effectively slow down or block the diffusion of hydrogen permeating into the crack tip [43], and decreases the EAC rate dramatically. Moreover, the growth of CPFs will also induce extra stress (referred to as CPF-induced stress), which could lead to the redistribution of stress [44,45,46,47,48] and hydrogen redistribution [27], and eventually affect the EAC susceptibility [49,50,51,52].

In this study, the formation of CPFs and the accompanying CPF-induced stress at a crack tip of 316L were investigated by establishing a two-dimensional numerical model including material diffusion and solid mechanics in COMSOL. The effects of CPF on the hydrogen at the crack tip were also studied.

## 2. Diffusion of Oxygen and Hydrogen in the Metal

### 2.1. Mass Balance Equation

The diffusion of oxygen and hydrogen in stainless steel is similar to the transfer of dilute substances in solution, and its diffusion is defined according to the conservation of mass on oxygen or hydrogen concentration,
(1)$$\frac{\partial }{\partial t}\int_{V}C_{i}dV+\int_{\partial V}J_{i}\cdot ndS=0,$$
where *∂*/*∂t* is the partial derivative of time; *V* is an arbitrary volume with surface area *S*, *n* denotes the direction of the outer normal of the volume *V*; *J_i_* denotes the flux of species *i* through the outer surface, mol·m^−2^·s^−1^; *C_i_* is the concentration of species *i*, m^2^·s^−1^, *i* denotes oxygen or hydrogen.

### 2.2. Diffusion Driving Force and Flux

The diffusion of oxygen and hydrogen in metals is a substance exchange process that continues until thermodynamic equilibrium is reached. Typically, diffusion may be caused by a variety of factors, such as chemical potential *μ* [53,54], temperature *T* [55], pressure *P* [56], and other external potentials, so the diffusion driving force [53,57,58] can be a gradient of one or more of these factors. When several of these processes occur simultaneously, the problem becomes too complex to solve, so it is considered that all driving forces can be reduced to a potential chemical gradient to simplify the problem. In this case, the mass fluxes [59] are
(2)$$J_{i}=-\frac{D_{i}}{RT}C_{i}\nabla \mu_{i},$$
where *D_i_* is the diffusion coefficient, m^2^·s^−1^; *R* is the gas constant, J·mol^−1^·K^−1^; *T* is absolute zero, K; *μ_i_* is the chemical potential.

In Equation (2), diffusion is controlled by the chemical potential gradient; this equation is traditionally expressed in terms of concentration. For dilute solutions, the chemical potential of oxygen or hydrogen (without considering the stress state) can be achieved by replacing the activity with their concentration as follows:(3)$$\mu_{i}=\mu_{i0}+RTlnC_{i},$$
where *μ_i_*_0_ is the chemical potential at the reference temperature and pressure of oxygen or hydrogen.

When a solid is stressed, the distance between lattice sites widens and is subjected to tensile stress [60], which can lead to a decrease in chemical potential. So, it can be found that for stressed solids under different thermodynamic conditions,
(4)$$\mu \left(\sigma_{h}\right)=\mu \left(\sigma_{h0}\right)-\sigma_{h}V_{i},$$
where *σ*_h_ is the hydrostatic stress, which is calculated by the diagonal terms of the stress tensor, *σ*_h_ = ∑*σ_j_*/3; *V_i_* is the partial molar volume of oxygen or hydrogen, m^3^·mol^−1^; *μ*(*σ*_h0_) is the chemical potential at the initial stress condition; *μ*(*σ*_h_) denotes the chemical potential with stress *σ*_h_. From Equation (4), it can be seen that the interaction of oxygen or hydrogen atoms with the stress field depends on the stress tensor *σ_j_* and the partial molar volume *V_i_*.

By combining Equations (3) and (4), the chemical potential of hydrogen in the metal under force can be described as
(5)$$\mu_{i}=\mu_{i0}+RT\ln C_{i}-\sigma_{h}V_{i},$$

By substituting Equation (5) into (2), the flux equation can be expressed as
(6)$$J_{i}=-D_{i}\nabla C_{i}+\frac{D_{i}}{RT}C_{i}V_{i}\nabla \sigma_{h},$$

### 2.3. Analogy of Diffusion Model in COMSOL

Substituting Equation (6) into (1) yields
(7)$$\frac{\partial }{\partial t}\int_{V}C_{i}dV+\int_{\partial V}\left(-D_{i}\nabla C_{i}+\frac{D_{i}}{RT}CV_{i}\nabla \sigma_{h}\right)\cdot ndS=0,$$

The simplified mass diffusion instanton equation is expressed as
(8)$$\frac{\partial C_{i}}{\partial t}+\nabla \left(-D_{i}\nabla C_{i}\right)+\nabla \left(\frac{D_{i}CV_{i}}{RT}\nabla \sigma_{h}\right)=0,$$

However, in COMSOL, the equation for the conservation of mass of a substance in a dilute solution is
(9)$$\frac{\partial C_{i}}{\partial t}+\nabla \left(-D_{i}\nabla C_{i}\right)+u_{i}\cdot \nabla C_{i}=R_{a},$$
where *R*_a_ is the reaction rate of species *i*, mol·m^−3^·s^−1^; *u_i_* is the convection velocity of species *i*, m·s^−1^.

The second term in Equation (9) is the diffusion term, which represents the interaction between the dilute substance and the solvent. The third term indicates the convective transport due to the velocity field. On the right side of the equation, *R_a_* denotes the source or adsorption term. To realize the equation for hydrogen transport under the coupling given in Equation (8), an analogy is drawn with Equation (9):(10)$$\frac{\partial C_{i}}{\partial t}+\nabla \left(-D_{i}\nabla C_{i}\right)+\frac{D_{i}V_{i}}{RT}\nabla \sigma_{h}\cdot \nabla C_{i}=-\frac{D_{i}V_{i}}{RT}\nabla^{2}\sigma_{h}C_{i},$$

By comparing Equations (9) and (10), the convection velocity *u* and reaction rate *R_a_* can be found as
(11)$$u_{i}=\frac{D_{i}V_{i}}{RT}\nabla \sigma_{h},$$
(12)$$R_{a}=-\frac{D_{i}V_{i}}{RT}\nabla^{2}\sigma_{h}C_{i},$$

Thus, the convection velocity *u_i_* is determined by the diffusion coefficient, the partial molar volume of species *i* and the hydrostatic pressure gradient in the solid. Assuming that no species is injected into or flowing out of the metal surface, the reaction rate *R*_a_ is considered to be the total rate of species adsorption or diffusion.

### 2.4. Boundary of the Effective Passivation Film

It is commonly believed that the passive film on the surface of stainless steel has a bilayer structure, an outer layer composed of iron oxides and hydroxides, and an enriched chromium oxide inner layer [61,62]. However, the structure and composition of the passive films vary with different testing environments and technologies. Even Hakiki et al. [63] have observed a distinct bilayer with a Cr-oxide inner layer and Fe-oxide outer layer on the surface of 304 and 316 SSs, Betova et al. [64] found that the passive film was composed of either a single (Fe, Cr)_2_O_3_ layer or a double layer (i.e., FeCr_2_O_4_ inner layer and (Fe, Cr)_2_O_3_ outer layer). As it is difficult to identify the actual structure of the passive film, the passive film is simplified as a single layer composed of Fe_2_O_3_, and the formation of the passive is controlled by the diffusion of oxygen at the crack tip. For a fully passivated film, the oxygen concentration could be calculated as
(13)$$c_{o}=3c_{\mathrm{film}},$$
(14)$$c_{\mathrm{film}}=\frac{\rho_{\mathrm{film}}}{M_{\mathrm{film}}},$$
where *c*_o_ is the molar concentration of oxygen at complete passivation, mol·cm^−3^; *c*_film_ is the molar concentration of fully passivated film, mol·cm^−3^; *ρ*_film_ is the film density, g·cm^−3^; *M*_film_ is the molar mass of the film, g·mol^−1^.

As it is only a fully passivated film that can block the permeation of hydrogen into the metal [65], the boundary of a fully passivated film should be determined. As the fully passivated layer only contains Fe_2_O_3_, the effective film region could be where the oxygen concentration is above the value of *c*_o_ calculated by Equation (13).

With a density of 5.24 g·cm^−3^ and a molecular weight of 159.69 for the Fe_2_O_3_, according to Equations (13) and (14), *c*_film_ and *c*_o_ are 3.28 × 10^4^ mol·m^−3^ and 9.84 × 10^4^ mol·m^−3^, respectively. It should be noted that these values are calculated with the parameters of Fe_2_O_3_, actually, the crack tip is composed of Fe initially, so the oxygen needed to transform Fe to Fe_2_O_3_ should be,
(15)$$c_{o}=\frac{3}{2}c_{\mathrm{Fe}},$$
(16)$$c_{\mathrm{Fe}}=\frac{\rho_{\mathrm{Fe}}}{M_{\mathrm{Fe}}},$$
where *c*_Fe_ is the molar concentration of fully passivated film, mol·m^−3^; *ρ*_Fe_ is the density of Fe, kg·m^−3^; *M*_Fe_ is the atomic mass of Fe.

By substituting the density of 7.87 g·cm^−3^ and an atomic mass of 55.86 for Fe into Equations (15) and (16), *c*_Fe_ and *c*_o_ are 1.41 × 10^5^ mol·m^−3^ and 2.11 × 10^5^ mol·m^−3^, respectively. However, when the oxygen is permeated into the crack tip, an extra expansion will occur at the crack tip, which will lead to the reduction in the molar concentration of Fe, so the oxygen concentration to determine the boundary of an effective film should be in the range between 9.84 × 10^4^ mol·m^−3^ and 2.11 × 10^5^ mol·m^−3^.

### 2.5. CPFs-Induced Stress

As the formation of the passivation film is a permeation of oxygen into the metal, the film-induced stress was assumed to be generated by the volume expansion when Fe transferred to Fe_2_O_3_. The process could be simulated by the hygroscopic swelling module for moisture concentration coupling between the solid mechanics and the transport of diluted species in COMSOL. As an internal strain caused by the change of moisture content, the strain induced by the hygroscopic swelling can be written as
(17)$$\varepsilon =\beta M_{o}\left(c_{\mathrm{mo}}-c_{\mathrm{mo},\mathrm{ref}}\right),$$
where *ε* is the strain caused by the absorbed oxygen; *β* is the hygroscopic expansion coefficient, m^3^·kg^−1^; *M*_o_ is the molar mass of oxygen permeated, and it equals 15.999 g·mol^−1^; *c*_mo_ is the molar concentration of oxygen permeated into the metal, mol·m^−3^; *c*_mo,ref_ is the strain-free reference concentration of oxygen in the metal, mol·m^−3^, and it is assumed to be 0 with no oxygen at the crack tip initially.

The hygroscopic expansion coefficient *β* during the formation of CPFs is calculated by
(18)$$\beta =\frac{\varepsilon_{v}}{M_{o}c_{o}},$$
(19)$$\varepsilon_{v}=\frac{v_{\mathrm{film}}-v_{\mathrm{Fe}}}{v_{\mathrm{Fe}}}=\frac{c_{\mathrm{Fe}}-c_{\mathrm{film}}}{c_{\mathrm{film}}}$$
where *ε*_v_ the volume strain when Fe is completely transformed to Fe_2_O_3_, *c*_o_ is the molar concentration of oxygen in a completely passivated film, mol·m^−3^.

Sturchio [66] mentioned in his paper that *ε*_v_ should be greater than 0.00035. According to Equations (13)–(15), (18) and (19), the hygroscopic expansion coefficient *β* could be calculated as should be in the range between 2.2 × 10^−7^ m^3^·kg^−1^ and 2.10 × 10^−3^ m^3^·kg^−1^.

## 3. Experiments and Finite Element Calculation

### 3.1. Tensile Test of 316L

According to the GB/T228.1-2010, the plate tensile specimens shown in Figure 1 were made from the as-received 316L austenitic stainless steel sheet of 400 mm × 400 mm × 2 mm by wire cutting and deburring; the chemical composition of 316L is shown in Table 1. The surface of the sample was polished with a surface roughness of less than 3.2 μm. The uniaxial tensile test was carried out at room temperature on the PLD-50 fatigue tensile tester made by Xi’an Lichuang Material Testing Technology Co., Ltd. (Xi’an, China), and the specimens were stretched to fracture under a velocity of 2 mm·min^−1^ to obtain the mechanical properties of 316L austenitic stainless steel. The obtained engineering stress–strain curve was converted to a real stress–strain curve, as shown in Figure 2.

### 3.2. Simulation Models

#### 3.2.1. Oxygen Diffusion Model

According to the geometric symmetry, a simplified half-plate specimen with a crevice located at the left side was adopted in the simulation, as shown in Figure 3. The length and width of the specimen were 2 *W* and *W*, respectively, and it was assumed that the crevice depth was *L* and had a blunt crack tip with a radius of *r*. The model was separated into the corrosive environment and the metal matrix. The model was meshed using free triangle elements with a refined region in the crack tip, as shown in Figure 3b,c.

#### 3.2.2. Hydrogen Diffusion Model

The geometry model of hydrogen diffusion is similar to the oxygen diffusion model shown in Figure 3, the only difference was the matrix at the crack tip in region 2. As shown in Figure 4, the CPFs were modeled in the crack tip to estimate their block effects on hydrogen diffusion. The thickness of CPFs was determined by the oxygen diffusion at the crack tip. According to Equations (13)–(16), the oxygen concentration in effective CPFs should be no less than 9.84 × 10^4^ mol·m^−3^ or 2.11 × 10^5^ mol·m^−3^ with different assumptions; however, the minimum value of 9.84 × 10^4^ mol·m^−3^ was selected arbitrarily to determine the thickness of CPFs in the simulation. The CPFs were modeled after the calculation in the oxygen diffusion model, and a much finer mesh was applied.

### 3.3. Boundary Conditions and Loads

During the simulation of oxygen diffusion, the crevice was assumed to be fulfilled with sodium chloride solution with a concentration of 0.01 mol·L^−1^ with an initial pH of 6.8. The boundary conditions and loads are shown in Figure 5. It was also assumed that there was no mass flux via all the edges. However, the crack tip surface was assumed to have a constant oxygen concentration of 9.84 × 10^4^ mol·m^−3^ when calculating the formation of corrosion production films; the value equaled to the dissolved oxygen in the solution. The displacement load was applied on the top edge in the range of 0~0.04 mm, which was intended to explore the effects of CPF-induced stress on the crack tip stress under different load conditions. The symmetrical boundary was used at the bottom edge.

The simulation environment in the hydrogen diffusion model was similar to the oxygen diffusion model. However, it was assumed that there was no hydrogen in the metal matrix initially, and a hydrogen concentration of 50 mol·m^3^ was applied on the crack tip surface as a boundary condition.

The relative parameters used in the simulation are listed in Table 2.

### 3.4. Simulation Procedures

The simulation was carried out in two stages. The first stage with three subprocesses was to consider the formation of CPFs and its effects on the crack tip stress distribution. Firstly, the stress distribution in the crack tip was studied under different displacement load conditions without considering the formation of CPFs; then, the formation of CPFs without applied displacement load; and finally, the formation of CPFs under different displacement conditions. The three subprocesses enable the observation of CPFs’ effects on crack tip stress distribution.

The second stage was hydrogen diffusion under different displacement load conditions without and with CPFs modeled to assess the block effects of CPFs on hydrogen diffusion.

## 4. Results and Discussion

### 4.1. Mesh Independence Verification

When establishing a finite element model, the main considerations are computational accuracy and computational scale, although increasing the number of meshes can better reflect the trend of data changes, the increase in the number of meshes will bring about a higher computational cost. So, a relatively coarse mesh was carried out at the crack tip region with a triangular mesh of 0.001 mm, while the other region was meshed by the adaptive meshing technique of the software, and a total of 25,813 meshes was obtained. The second mesh strategy used the two regular refinements carried out automatically by the software’s refinement scheme, and the total meshes are 40,707.

The Mises stress along the crack contour was plotted in Figure 6 to compare the computational errors of the two mesh strategies. The refined mesh which was increased by 57.7% has the same stress distribution as the coarse mesh. The result indicates that a further refined mesh has no obvious improvement on the results, so there is no need to carry out a more refined mesh so that the model keeps computational efficiency.

### 4.2. The Oxygen Diffusion and Formation of CPFs

Figure 7a shows the oxygen concentration distribution at the crack tip of stainless steel, in which the black contour line represents the boundary with minimum oxygen concentration to form an effective CPF, and the effective CPF boundary was extracted and redrew as in Figure 7b. By measuring the thickness of CPFs, a nonuniform distribution of 3 nm at the side of the crack (*θ* = 90°) and 2 nm at the crack tip (*θ* = 90°) was found, which could be caused by a larger contact surface with a corrosive environment at the side than the tip. The measured CPF boundary curve was used in the hydrogen diffusion model to build the film layer.

### 4.3. Crack Tip Stress Distribution

#### 4.3.1. CPF-Induced Stress

When the oxygen permeates into the metal matrix and reacts to form the CPFs, the CPF-induced stress emerges due to the volume expansion of materials. Figure 8 demonstrates the CPF-induced stress near the crack tip after oxygen diffusion without external loads applied. The formation of CPFs induced a high stress gradient close to the crack surface; the highest stress is more than 200 MPa. With a decreased oxygen concentration far away from the crack surface, the volume expansion induced by the formation of CPFs is suppressed, and thus the CPF-induced stress plunges to below 40 MPa within 0.2 mm. However, the highest CPF-induced stress region does not appear at the crack tip but locates approximately in the region of 50~60°, and this could be the interaction results of oxygen diffusion, stress–strain coordination, and the restraint conditions at the crack tip.

According to Equation (17), in a volumetric expansion induced by oxygen diffusion, the volumetric strain is always a positive correlation with oxygen concentration. Figure 9a compares the oxygen concentration and volumetric strain along the crack contour, in which the oxygen concentration has the lowest value approximately located at 60~70° and increases with an increased angle towards the crack wall or a decreased angle towards the crack tip, and following the highest oxygen concentration at the crack tip. The volumetric strain has a similar distribution trend with oxygen distribution along the crack contour according to the positive correlation relationship. However, the crack tip constraint conditions make the two curves somewhat different. Even the highest oxygen concentration locates at the crack tip, the largest volumetric strain appears near the crack wall with a relatively low oxygen concentration compared to the crack tip, and the smallest volumetric locates in the region of 50~60°, which has a shift compared to the lowest oxygen concentration region. The possible explanation should be the strict constraints at the crack tip which suppresses the volumetric expansion and leads to a small volumetric strain, while the loose constraint near the crack wall enables the volumetric expansion there and leads to a large volumetric strain.

Commonly, the stress induced by the volumetric expansion will increase if there is a strict constraint to suppress the expansion, and the stress increase will decrease if the constraint is loose. The volumetric expansion-induced stress shown in Figure 9b meets the law, the Mises stress in the crack tip (*θ* = 0°) is higher due to a suppressed volumetric expansion by the constraint, and the stress near the crack wall (*θ* = 90°) is low due to a loose constrained volumetric expansion. It should be noticed that in the crack contour of 50~60°, both the oxygen concentration and volumetric expansion are small, and so should the stress be according to the positive relationship between oxygen concentration and stress. However, the highest stress appears in this range, and it is explained by the volumetric expansion beside this region. With a low volumetric expansion in 50~60°, a low constraint appears here, thus the high volumetric expansion induced by the oxygen diffusion outside will expand towards it and result in high compressive stress accompanied by low oxygen concentration and low strain in the area.

#### 4.3.2. Stress Redistribution under External Tensile Loads

Figure 10 compares the Mises stress distribution under different load conditions before the formation of CPFs. When the tensile displacement increased from 0.01 mm to 0.04 mm, the maximum Mises stress at the crack tip increased with the high-stress zone increases.

Figure 11 compares the Mises stress distribution under different load conditions after the formation of CPFs. The maximum Mises stress and the high-stress zone also increase with the increased tensile displacement. By comparing Figure 10 and Figure 11, the formation of CPFs will decrease the maximum Mises stress at the crack tip at the low tensile displacement of 0.01 mm and 0.02 mm, while the maximum Mises stress seems the same at the high-tensile displacement stage. The results indicate that the formation of CPFs has more observed effects on crack tip stress.

Figure 12 plots 200 MPa Mises stress isolines before and after the formation of CPFs; the increased isoline zones in Figure 12a,b indicate the crack tip Mises stress has a positive correlative relationship with the tensile displacement load, while the 200 MPa contours shrink significantly after the formation of CPFs, which shows the weakening effects of CPF-induced stress on crack tip Mises stress.

A detailed effect of the CPF formation on Mises stress distribution near the crack tip is shown in Figure 13, which compares the Mises stress distribution change after the formation of CPFs. As shown in Figure 13a, when an external tensile displacement load of 0.01 mm exists, the formation of CPFs significantly reduces the crack tip Mises stress by about 115 MPa, and the reduction effects decrease with an increasing distance from the crack tip. After the distance increases to a critical position of about 0.1 mm, the CPF-induced stress tends to increase the Mises stress, and the CPF formation could have an enhancement of 30~40 MPa on Mises stress with the distance increases to 0.2 mm~0.5 mm. The increasing external tensile displacement load tends to reduce the CPF formation on the crack tip, as the Mises stress difference narrows to only 11.7 MPa with an increased tensile displacement load of 0.04 mm. However, the Mises stress reduction region caused by CPFs enlarges with an increased tensile displacement load, as the critical position, at where the CPFs effects on Mises stress change from reduction to enhancement, increases from 0.1 mm to 0.21 mm, and simultaneously, the CPF enhancement on Mises stress far from the crack tip decreases slightly with the increase in tensile displacement.

The CPFs also have weakening or enhancement effects on Mises stress along the crack contour. As shown in Figure 13b, with an external tensile displacement of 0.01 mm, the Mises stress has the largest reduction of 115 MPa induced by CPFs, and the value decreases with an increasing angle towards the crack wall. The critical angle is about 72°, after which the CPFs will enhance the Mises stress, and the enhancement could be up to 73 MPa at 90°. However, the stress difference curves are much close to 0 with an increased external tensile displacement, which indicates that the effects of CPFs on crack tip stress decrease with an external tensile displacement, no matter the enhancement or weakening effects.

The CPFs weakening on Mises stresses very close to the crack tip could be contributed to the opposite stress direction caused by external loads and CPF-induced stress. Figure 14 shows the distribution of hydrostatic pressure induced by CPFs and external loads, which can character the stress state of compress or tensile and acts to change the volume of a material element. The external displacement loads always induce a negative hydrostatic pressure both in front of the crack tip and around the crack contour and will have behavior tensile effects on the material, while the formation of CPFs without external loads will always induce a positive hydrostatic pressure, which has a compressed effect on the material. The compress effects induced by CPFs weaken the tensile stress caused by external loads and thus decrease the crack tip stress. When the external displacement is 0.01 mm, the hydrostatic press caused by external loads is close to the values induced by CPFs, so the CPF weakening effects on the crack tip Mises stress is significant. However, the stress induced by CPFs is relatively small compared to an increased external displacement load, and thus its effects on crack tip Mises stress decreases.

With an increased external displacement load, the hydrostatic pressure along the crack contour increases and decreases from the crack tip towards the crack wall. At small external load conditions, the hydrostatic pressure has the highest value at the crack tip and decreases when far away from it. However, when the load exceeds a critical value, the hydrostatic pressure curves will change to a convex function, and the highest hydrostatic pressure will move from the crack tip inside to the matrix. The larger the external displacement, the higher the highest hydrostatic pressure, and the more insider the position.

The CPF formation efforts on hydrostatic pressure distribution are shown in Figure 15. Negative hydrostatic pressures still exist in front of the crack tip, and the stress status is still tensile, but their values are decreased compared to Figure 14, while along the crack contour, the hydrostatic pressure curves also have a positive movement, which results in a whole decrease in tensile status, and the stress status changes from tensile to compress when the angle exceeds 38° with an external displacement equal to 0.01 mm.

### 4.4. The Hydrogen Diffusion at the Crack Tip

Figure 16 demonstrates the significant inhibition effects of CPF formation on the hydrogen permeation into the material. With the hydrogen influx into the stainless steel matrix, the concentration near the crack tip surface is up to 49 mol·m^−3^ and reduces to a relatively high level of 30 mol·m^−3^ in about 0.5 mm if there are no CPFs. While the hydrogen concentration declines to 14 mol·m^−3^ near the crack tip surface and decreases to about 8 mol·m^−3^ far from it when the crack surface is covered by CPFs. The CPFs have dramatically blocked the permeation of hydrogen due to the low hydrogen diffusion coefficient in CPFs.

Figure 17 shows the hydrogen distribution without CPF formation under different load conditions. When there is no displacement load, the hydrogen concentration gradient is the only driving force of hydrogen diffusion in the matrix, so the hydrogen has the highest concentration at the crack tip, and decreases when deepening into the matrix, while the hydrogen concentration has little differences along the crack contour. If there are any loads applied, the hydrostatic pressure gradient will drive the hydrogen to converge to a high-pressure region. With a small displacement load of 0.01 mm, the hydrogen distribution in front of the crack tip is almost unchanged when applying a displacement load of 0.01 mm. However, a slightly increased hydrogen concentration at the crack tip and a decreased hydrogen concentration far from it could be found, and this is due to the increased hydrostatic pressure at the crack tip and decreased hydrostatic pressure far from it, which provides a converse diffusion direction with the hydrogen concentration gradient. When the external displacement loads increase to 0.03 mm or 0.04 mm, the hydrostatic pressure distribution curves shown in Figure 14a change to a convex function, which will enhance the hydrogen diffuses to the extreme point region of the curve; thus, the highest hydrogen concentration appears in this region. The larger the external displacement load, the higher the highest hydrogen concentration appears. Figure 17b illustrates that the hydrogen concentration differences are small along the crack contour. With an increased external displacement load, the hydrogen concentration increases slightly in the tip, and the high hydrogen concentration region extends from the crack tip towards the crack wall.

Figure 18 shows the effects of CPF formation on hydrogen distribution under different load conditions. Considering the hydrogen distribution in front of the crack tip, the hydrogen concentration has the highest value at the crack tip without an external displacement load applied and decreases with an increased distance from it. When an external displacement of 0.01 mm is applied, the hydrogen concentration decreases, and this could be explained by the hydrostatic pressure change. Due to the opposite tensile stress effect of the external load from the compression effects of CPF-induced stress on the crack tip, the CPF-induced stress is unloaded when applying a tensile displacement load of 0.01 mm, resulting in the reduction in the overall stress level and hydrostatic pressure, thereby weakening the diffusion of hydrogen towards the crack tip from the low hydrostatic pressure area. However, when the external load increases to 0.02 mm and higher, the overall hydrogen concentration level begins to increase again, which is due to the joint effects of the external tensile load and the CPF-induced compressive stress, so that the crack tip is in a higher tensile stress state than the original CPF-induced compressive stress and increased hydrostatic pressure. Thus, there exists a tensile load threshold which could balance the CPF-induced compressive stress, and at this stage, the hydrogen is the lowest at the crack tip. A greater or smaller external tensile load than the threshold will lead to a higher stress status, which will drive the hydrogen converging to this region.

Different from the hydrogen distribution without CPF formation shown in Figure 17a, in which the highest hydrogen concentration appears inside the matrix at a high external level, the hydrogen shown in Figure 18a always has the highest hydrogen concentration at the crack tip and decreases with an increased distance from it after the formation of CPFs. Even the hydrostatic pressure curves shown in Figure 15a are convex functions when the displacement equals 0.03 mm or 0.04 mm; the highest hydrogen concentration does not appear in the extreme point region of the curve. Moreover, the hydrogen concentration at the crack tip increases from 12 mol·m^−3^ to 32 mol·m^−3^ with the external loads increasing from 0.01 mm to 0.04 mm when CPFs exist, while the hydrogen concentration change is less than 0.5 mol·m^−3^ without CPFs. The dramatic differences could be the combination of the CPFs’ suppression on hydrogen permeating into the matrix and the stress-driven hydrogen diffusion.

Due to the low diffusion coefficient in CPFs, it is difficult for the hydrogen to permeate through the CPFs into the matrix, thus the hydrogen concentration could mainly be affected by the stress-induced redistribution of hydrogen previously penetrated into the matrix. As shown in Figure 18b, when the external displacement increases to 0.01 mm, the crack tip hydrogen decreases due to the weakened effects of external loads on film-induced stress. With an increased external load, the hydrostatic pressure stress at the crack tip shown in Figure 15b has a more significant increase than the crack wall, which causes a high-hydrostatic pressure gradient to drive the hydrogen diffusing from the crack wall towards the crack tip, and results in a hydrogen loss at the crack wall and hydrogen enrichment at the crack tip. However, if there is no CPF formation, it is easy for hydrogen to permeate into the matrix, the hydrogen loss at the crack wall could be sufficiently supplied by the hydrogen influx; thus, an obvious difference could be found between the hydrogen distributions along the crack surface at different external load level without CPFs shown in Figure 17b.

## 5. Conclusions

A numerical model combining mechanical and diffusion fields was established to simulate the formation of CPFs and the diffusion of hydrogen. By establishing a simplified half-plate specimen with a crevice located at the left side in COMSOL, the formation of CPFs at the crack tip and its effects on the crack tip stress status and hydrogen diffusion were studied. The main conclusions can be summarized as follows:(1)By assuming a main composition of Fe_2_O_3_, the oxygen concentration needed to form effective CPFs is in the range between 9.84 × 10^4^ mol·m^−3^ and 2.11 × 10^5^ mol·m^−3^. The CPF-induced stress was determined by the hygroscopic swelling, and the calculated hygroscopic expansion coefficient is in the range between 2.2 × 10^−7^ m^3^·kg^−1^ and 2.10 × 10^−3^ m^3^·kg^−1^.(2)Nonuniform CPFs with 2~3 nm thickness are formed according to the oxygen diffusion, which induces a high compressive stress and a high stress gradient close to the crack surface. Due to the crack tip constraint condition, the highest CPF-induced stress over 200 MPa accompanying the lowest volumetric expansion approximately locates in the crack contour of 50~60°, while the position near the crack wall has the largest strain and smallest stress.(3)The CPF-induced stress has a weakening effect on the stress status of a tensile crack tip. However, the weakening effect decreases with an increased external tensile load.(4)Without CPFs, the hydrogen is almost unchanged along the crack contour with different external load conditions; while in front of the crack tip, the highest hydrogen concentration locates at the crack tip, and decreases with an increased distance from it at low external tensile load, but the largest hydrogen concentration location moves from the crack tip to the inside of the matrix with an increased external tensile load.(5)The formed CPFs significantly inhibit the hydrogen permeation into the crack tip, and the stress plays an important role in the distribution of permeated hydrogen. The hydrogen concentration always decreases with an increased distance from the crack tip under different external tensile loads, while along the crack contour, the hydrogen concentration at the crack tip is smaller than the crack wall at a low external tensile stage, but it increases to be the largest with an increased external tensile stress.

## Figures and Tables

**Figure 1 materials-16-05799-f001:**
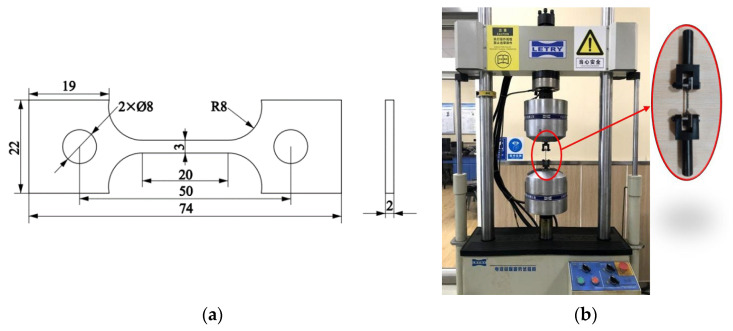
Uniaxial tensile test. (**a**) Geometric of the plate tensile specimen (mm); (**b**) specimen installation.

**Figure 2 materials-16-05799-f002:**
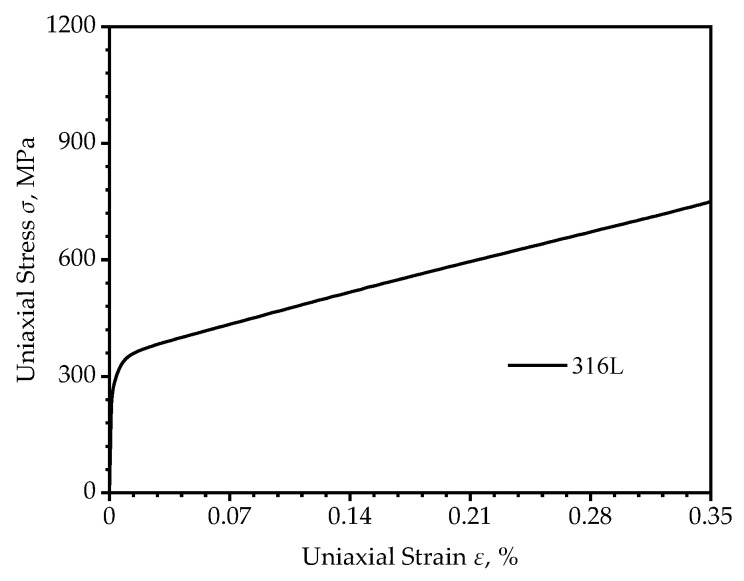
Real stress–strain curve of 316L.

**Figure 3 materials-16-05799-f003:**
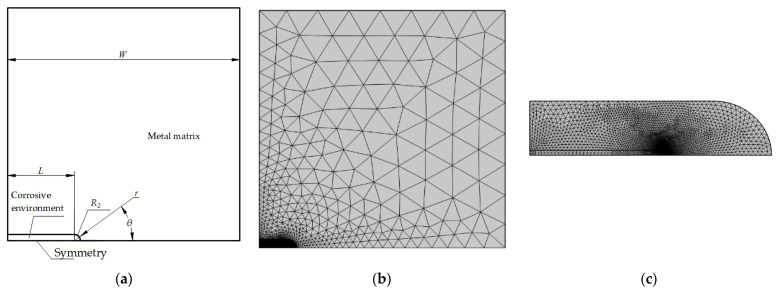
Geometry and mesh of calculation model. (**a**) Geometric dimension schematic; (**b**) overall meshing; (**c**) refined mesh region.

**Figure 4 materials-16-05799-f004:**
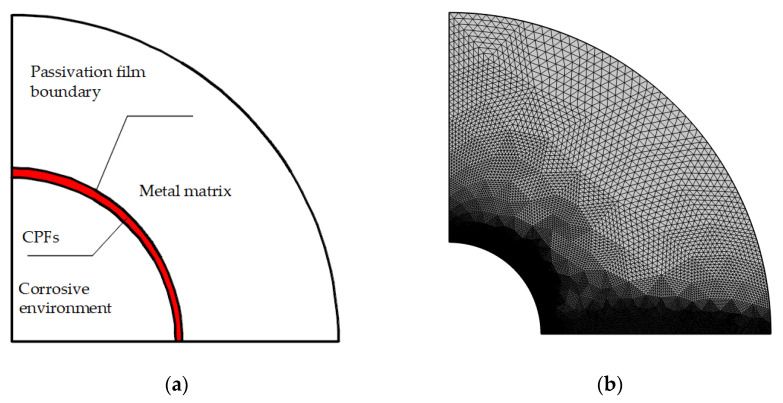
Geometry and mesh of crack tip with CPFs: (**a**) effective CPFs, (**b**) crack tip mesh.

**Figure 5 materials-16-05799-f005:**
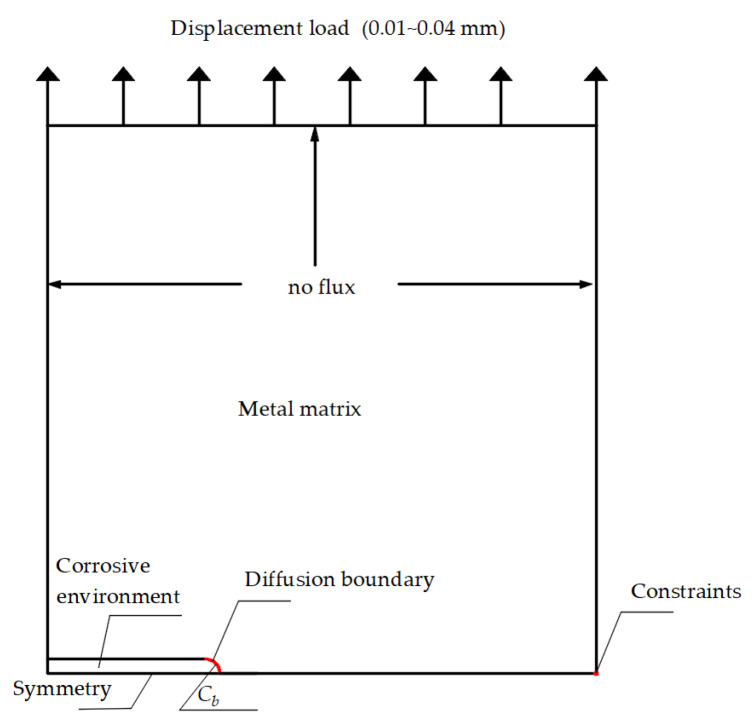
Boundary conditions and load applied, *C*_b_ represents the constant oxygen concentration or hydrogen concentration.

**Figure 6 materials-16-05799-f006:**
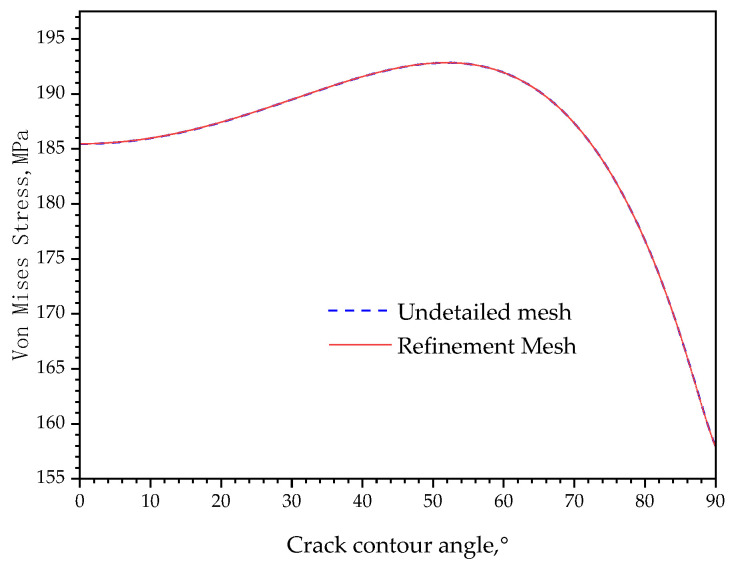
Stress comparison before and after mesh refinement.

**Figure 7 materials-16-05799-f007:**
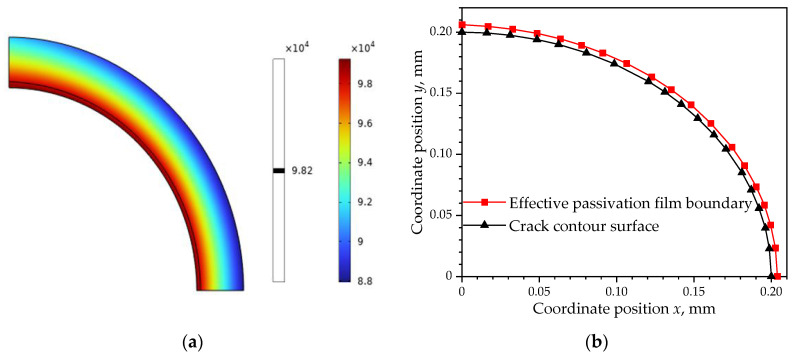
Oxygen concentration distribution at the crack tip: (**a**) concentration contour for effective CPFs, mol·m^−3^; (**b**) CPF thickness around the crack tip.

**Figure 8 materials-16-05799-f008:**
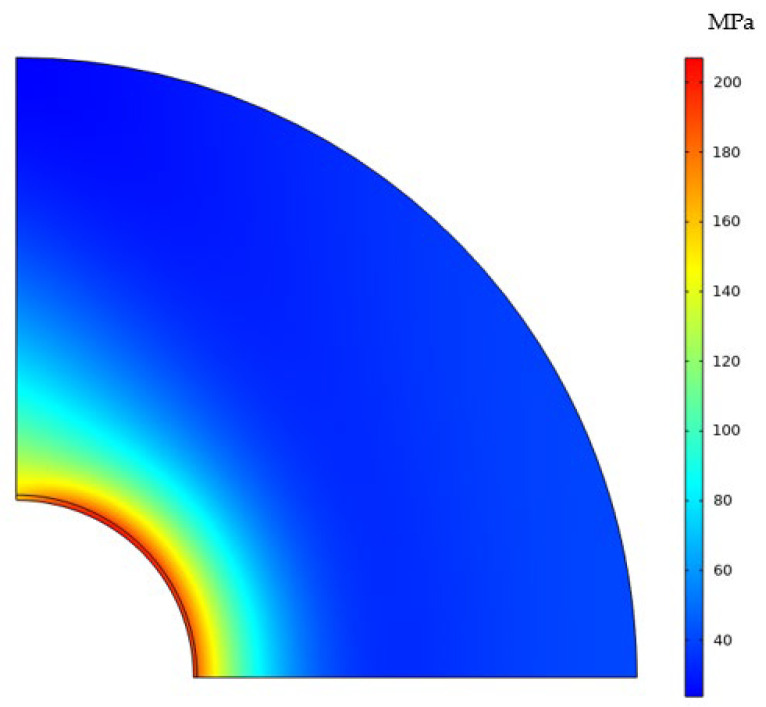
Stress distribution in the membrane without displacement load.

**Figure 9 materials-16-05799-f009:**
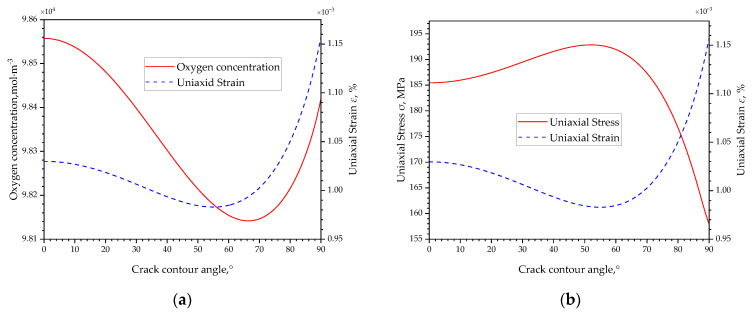
Stress, volume strain, and oxygen concentration along the crack contour. (**a**) Oxygen concentration and volume strain; (**b**) stress and volume strain.

**Figure 10 materials-16-05799-f010:**
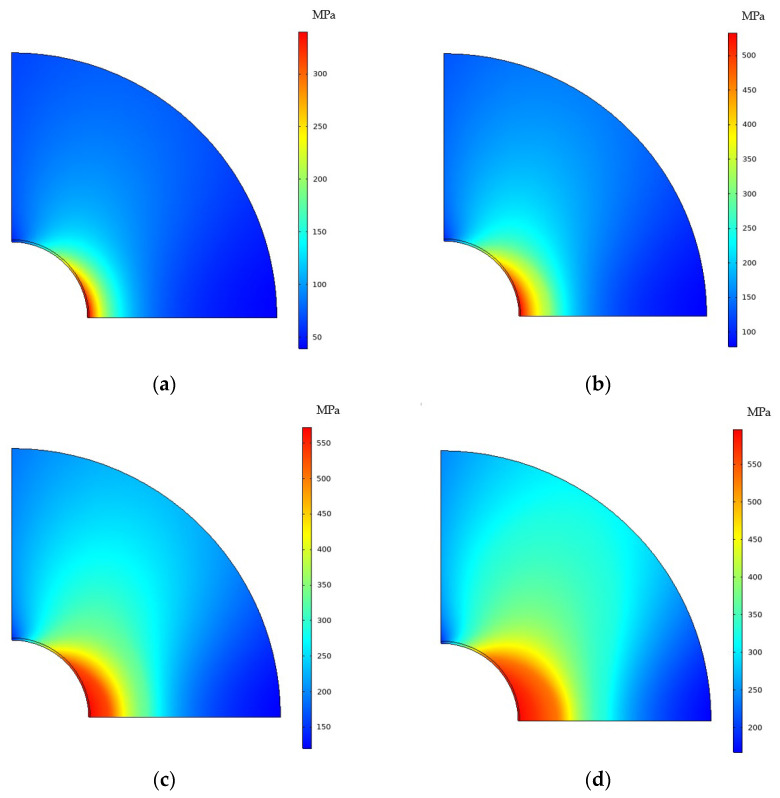
Stress distribution of different loads before the formation of CPFs: (**a**) *d* = 0.01 mm; (**b**) *d* = 0.02 mm; (**c**) *d* = 0.03 mm; (**d**) *d* = 0.04 mm.

**Figure 11 materials-16-05799-f011:**
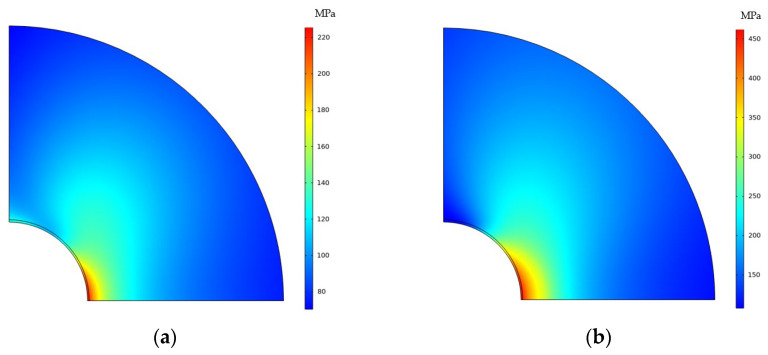
Stress distribution for different loads after the formation of CPFs: (**a**) *d* = 0.01 mm; (**b**) *d* = 0.02 mm; (**c**) *d* = 0.03 mm; (**d**) *d* = 0.04 mm.

**Figure 12 materials-16-05799-f012:**
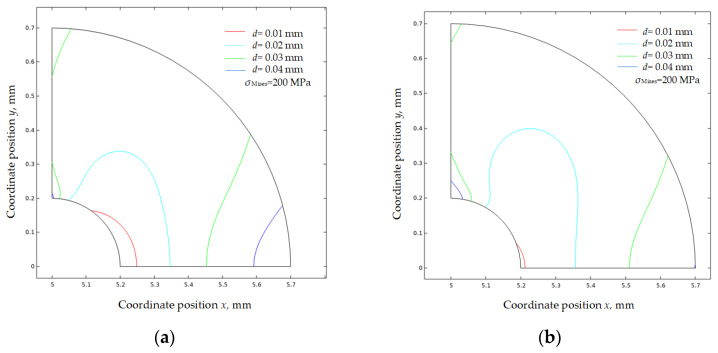
Mises Stress distribution near the crack tip: (**a**) before CPFs formation; (**b**) after CPF formation.

**Figure 13 materials-16-05799-f013:**
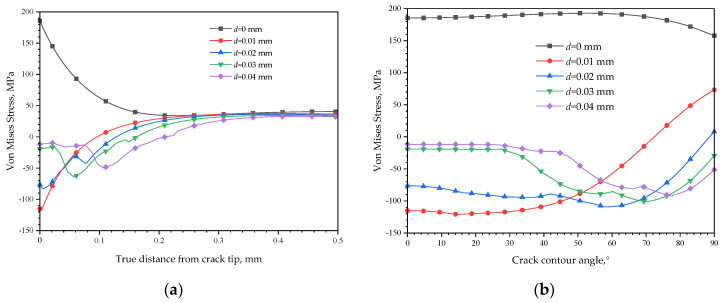
Mises stress distribution changes after the formation of CPFs: (**a**) in front of the crack tip; (**b**) along the crack contour.

**Figure 14 materials-16-05799-f014:**
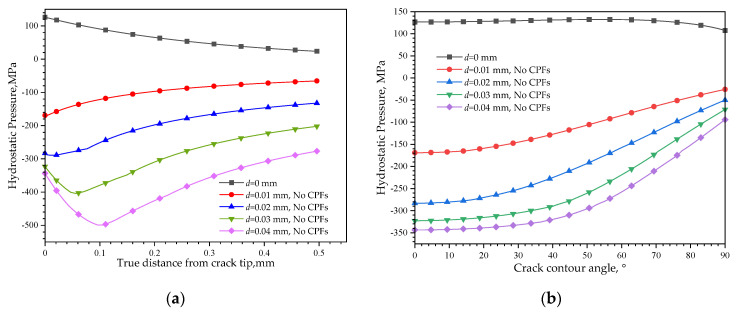
Hydrostatic pressure caused by CPFs formation or external loads: (**a**) in front of the crack tip; (**b**) along the crack contour.

**Figure 15 materials-16-05799-f015:**
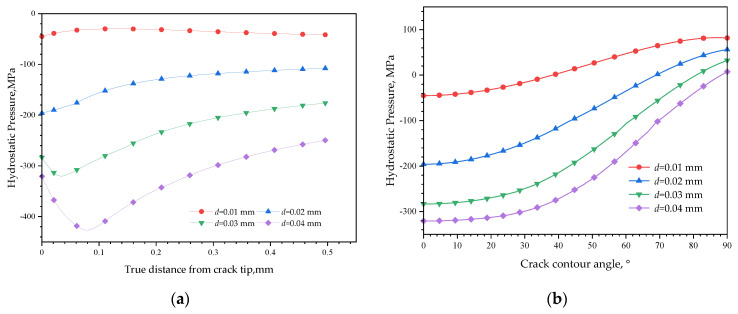
Hydrostatic pressure distribution after CPFs formation: (**a**) in front of the crack tip; (**b**) along the crack contour.

**Figure 16 materials-16-05799-f016:**
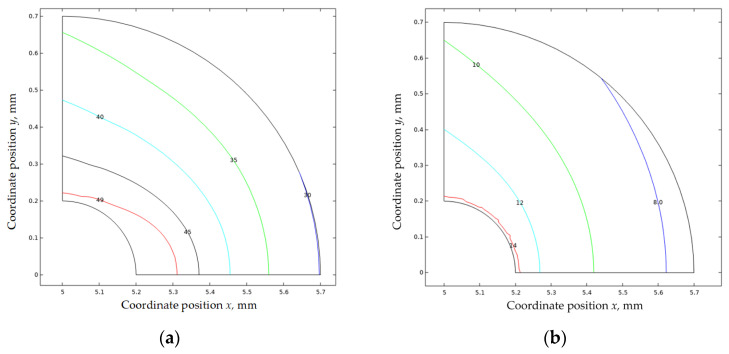
Distribution of hydrogen concentration in the crack tip region of stainless steel: (**a**) without CPFs; (**b**) with CPFs.

**Figure 17 materials-16-05799-f017:**
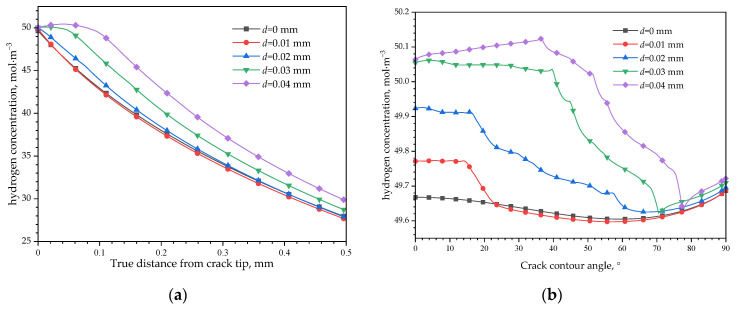
Hydrogen concentration distribution before CPFs formation: (**a**) in front of crack tip; (**b**) along crack contour.

**Figure 18 materials-16-05799-f018:**
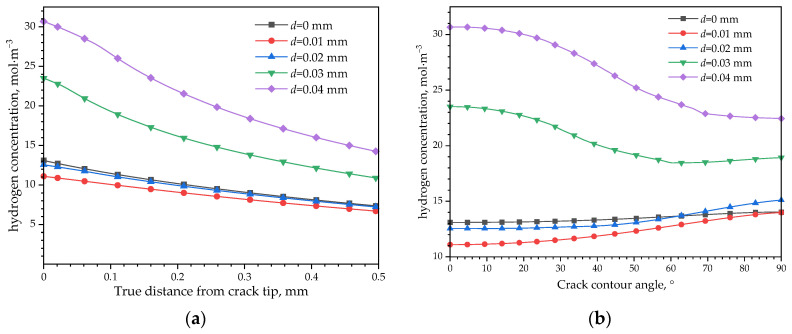
Hydrogen concentration distribution after CPF formation. (**a**) In front of the crack tip; (**b**) along the crack contour.

**Table 1 materials-16-05799-t001:** The 316L austenitic stainless steel main chemical composition (mass fraction).

Composition	C	Si	Mn	P	S	Cr	Ni	Mo
Content	0.030	0.370	1.300	0.026	0.029	17.050	10.230	2.100

**Table 2 materials-16-05799-t002:** Geometry and physical parameters used in the simulation.

Physical Parameters	Value
Plate width *W*, [mm]	60
Crevice depth *L*, [mm]	5
Crack tip blunt radius of *r*, [mm]	0.2
Metal density *ρ*, [kg·m^−3^]	7980
Elasticity Modulus *E*, [GPa]	176
Poisson’s ratio *v*	0.3
Initial yield stress, [MPa]	309
Oxygen diffusion coefficient *D*_O2_, [cm^2^·s^−1^]	2.29 × 10^−5^
Hydrogen diffusion coefficient in metal *D*_H_m_, [mm^2^·s^−1^]	3.3 × 10^−9^
Hydrogen diffusion coefficient in CPFs *D*_H_f_, [mm^2^·s^−1^]	7.57 × 10^−14^
Paramolar volume *V*_m_, [m^3^·mol^−1^]	7.13 × 10^−6^
Faraday’s constant *F*, [C·mol^−1^]	96,500
Absolute temperature *T*, [K]	298.15
Gas constants *R*, [mol·K·J^−1^]	8.314

## Data Availability

The data used to support the findings of this study were calculated according to the finite element method, and they are included in the article. The parameters used in the calculation model were cited from the references listed.

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
