# Peer review of "Modeling Corrosion Product Film Formation and Hydrogen Diffusion at the Crack Tip of Austenitic Stainless Steel"

_materials, 2023, doi:10.3390/ma16175799_

Round 1

Reviewer 1 Report

The article is interesting for researchers.

However, there are several questions for understanding the presented article.

1. Is it possible to use this model for other materials?

2. Are the simulation results comparable with the experimental results?

3. What structural changes correspond to one or another stage in the modeling?

Author Response

See annex for response materials.

Reviewer 2 Report

The article deals with modeling the corrosion film formation in an AISI 316 stainless steel. It is physics-based, and the resulting model is numerically tested. The paper is organized, detailed, and well-written. Some numerical details need to be clarified. Some issues to be addressed before considering for publication:

Some minor details to be addressed:

Figs 4 and 8 should be enhanced to show the stress gradient in more detail. Perhaps a contour plot rather than a discrete plot that you actually have.

In Eq (4), is the Vi partial molar volume present in the media the component is submerged in, or is the partial molar volume already diffused on the component?

Major observations:

In the literature review, the authors left other modeling techniques, such as parametric models, data driven-models, and other numerical methods, such as the BEM, See Xu et al. (2023) below

Add UTS maker and cell load used. Add all the experimental details.

Did the authors consider the state of oxidation? (Line 150). Steel corrosion can form several oxides byproducts (see Nasrazardani, 2007). They documented how the iron byproducts changes over time; hence Oxygen and hydrogen content change over time.

Did you consider the crack not being fully closed by the trapped corrosion byproduct?

For the sake of discussion and comparison, I recommend the authors add the SS316 chemical composition in molar concentration.

Did you simulate just one case? I do not understand how the boundary conditions were applied. A plot of boundary conditions might be helpful. It is difficult to assess results, let alone reproduce results without clear boundary conditions.

How do you validate results? Did you perform a mesh independence study?

Mentioned references:

Nasrazadani, et al. Effects of DBU, morpholine, and DMA on corrosion of low carbon steel exposed to steam, Corrosion Science, (49) 7. 2007, Pages 3024-3039,

https://doi.org/10.1016/j.corsci.2007.01.012.

Xu D, et al.  A Review of Trends in Corrosion-Resistant Structural Steels Research-From Theoretical Simulation to Data-Driven Directions. Materials (Basel). 2023 Apr 26;16(9):3396. doi: 10.3390/ma16093396

A thorough  language revision  should be performed

In the sense of force over area, pressure is singular. It is a noncountable noun, so is never plural.

L58 change  it’s for it is

L69 change government for governing

L125 revise the whole line

L452 change  combined for combining

Author Response

See annex for response materials.

Round 2

Reviewer 2 Report

The authors have provided satisfactory answers.  I recommend The paper be accepted upon checking minor details.

Good work. Hope to see more along that line.

L273 & L275. The appropriate term might be nodes or elements instead of meshes. Please check.